# Is Bilirubin Associated with the Severity of Ischemic Stroke? A Dose Response Meta-Analysis

**DOI:** 10.3390/jcm11123262

**Published:** 2022-06-07

**Authors:** Yumeng Song, Xiaohong Zhang, Chaoxiu Li, Shuang Xu, Baosen Zhou, Xiaomei Wu

**Affiliations:** 1Department of Clinical Epidemiology and Center of Evidence Based Medicine, The First Hospital of China Medical University, Shenyang 110000, China; 2020120211@stu.cmu.edu.cn (Y.S.); 2020120213@stu.cmu.edu.cn (C.L.); 2Department of Clinical Epidemiology, The Fourth Hospital of China Medical University, Shenyang 110000, China; 20052202@cmu.edu.cn; 3Technology Service Department, Library of China Medical University, Shenyang 110000, China; sxu@cmu.edu.cn

**Keywords:** bilirubin, stroke severity, ischemic stroke, oxidative stress

## Abstract

There is no consensus on the role of bilirubin in acute ischemic stroke. Higher levels of serum bilirubin may provide a treatment advantage in oxidative-stress-mediated diseases but also may simply reflect the strength of the oxidative stress. As of 28 February 2022, the relevant studies were selected from four databases (PubMed, Web of science, Cochrane, and CNKI) through a retrieval strategy, and strict literature screening and quality evaluation were carried out. The dose–response relationship was fitted with a restricted cubic splines function. We found that the serum total bilirubin level and the direct bilirubin level were positively correlated with the severity of ischemic stroke. The direct bilirubin level was linearly correlated with the severity of stroke (P for non-linearity = 0.55), and the direct bilirubin increase of 1 μmol/L may be related to the 1% increase in the possibility of having moderate or severe ischemic stroke. High bilirubin levels are associated with stroke severity in patients with ischemic stroke and may serve as a marker of the intensity of initial oxidative stress.

## 1. Introduction

Stroke is a clinical syndrome of a focal neurological defect and has been considered to be the second leading cause of death around the world and an important cause of disability in both developed and developing countries [1]. Ischemic stroke accounts for almost 70% of all stroke cases [2]. In all types of ischemic stroke, excessive oxidative stress leads to the structural and functional impairment of the brain, especially throughout the acute phase, and oxidative stress may play an important role in the pathophysiology of brain injury [3]. Serum bilirubin has been considered as the product of heme catabolism, and it has been widely recognized as an important biochemical indicator for the diagnosis of liver, bile, and blood system diseases [4], and a powerful physiological antioxidant [5]. Herishanu et al. [6] found a higher rate of hyperbilirubinemia in acute ischemic stroke, but they only found this phenomenon and did not explain it in detail. Luo [7] and Pineda et al. [8] have indicated that more severe strokes are associated with higher levels of oxidative stress and that increased serum bilirubin may reflect a systemic oxidative stress response after stroke. However, it remains unclear whether bilirubin plays a protective role in the pathophysiology of acute ischemic stroke brain injury or just reflects the severity of oxidative stress, or both [5]. At the same time, there has been no detailed assessment of the dose–response relationship between the two. Therefore, it is essential to systematically evaluate the correlation between circulating bilirubin and stroke. In this study, a meta-analysis was performed to explore the association between bilirubin, including direct and indirect bilirubin, and acute ischemic stroke (AIS) severity in patients after the occurrence of ischemic stroke, to provide a reference for investigating whether serum bilirubin can be an assessment indicator of ischemic stroke severity.

## 2. Materials and Methods

This systematic review was conducted according to the guidelines of Preferred Reporting Items for Systematic Reviews and Meta-Analyses (PRISMA) [9] (Appendix A).

### 2.1. Search Strategy

Two reviewers (Yumeng Song and Xiaomei Wu) searched four databases—PubMed, Cochrane Central Register of Controlled Trials (CENTRAL), Web of Science, and China National Knowledge Infrastructure (CNKI)—for the study on the relationship between serum bilirubin level and stroke before 1 March 2022. In order to collect as many relevant studies as possible, combine search related MeSH terms such as “Bilirubin” [Mesh], “Ischemic Stroke [MeSH]”, etc. (Appendix A); no language or study type restrictions were applied. To ascertain that the retrieved literature was sufficiently comprehensive, we also searched the reference lists of all relevant articles.

### 2.2. Study Selection

We used a standardized table by Microsoft Excel (Microsoft Excel software, version 2016, Microsoft Corporation, Redmond, WA, USA) The title and abstract of the initially retrieved literature were first preliminarily screened, and then all potentially relevant articles were evaluated based on the full text. The criteria were:The article reported the relationship between bilirubin level and ischemic stroke severity, or the available data were provided to calculate the corresponding estimates.Eligible study types include cohort studies, case-control studies, and cross-sectional studies that are also original human studies.If more than two articles came from the same research population, the latest or highest-quality result was adopted. If the article does not meet the above criteria, it is not considered.

All differences in inclusion studies were adjudicated by the authors.

### 2.3. Data Extraction

Two reviewers (Yumeng Song and Xiaohong Zhang) independently evaluated and extracted data, double-checked the available data, and completed the table. The following data were extracted: first author; publication year; country; design; subject source; mean or median bilirubin and standard deviation (SD) of bilirubin; number of participants/cases; and effect estimates and 95% confidence intervals (CIs) of study outcomes across categories of bilirubin, baseline characteristics of sample population (age, body mass index (BMI), smokers, alcohol, disease status such as hypertension, diabetes, hyperlipidemia), included and excluded criteria of original studies, and other factors. The result of the pooled estimate was generated using the adjusted effect estimate and the baseline measure of bilirubin exposure. For studies that presented several estimates adjusted for different numbers of potential confounders, the estimate that was adjusted for the highest number of potential confounders was selected for analysis. In order to obtain the standard effect value of bilirubin, we unified the unit of bilirubin; if the unit of bilirubin level was mg/dL, we multiplied it by 17.1 to convert it to μmol/L.

### 2.4. Literature Quality Assessment

Two authors (Xiaohong Zhang and Yumeng Song) independently assessed the quality of the included studies. Study quality was evaluated using the Newcastle–Ottawa Scale (NOS) for cohort studies and case-control studies [10] or the NOS modified for cross-sectional studies [11]. The quality of studies was determined according to the selection criteria for participants, the comparability of cases and controls, and the exposure and outcome assessments. A score of 7 or more was regarded as “high-quality”; otherwise, the study was regarded as “low-quality”. Over all, a score ≥5 indicated adequate quality for inclusion in the present review.

### 2.5. Outcome Definition

Ischemic stroke was defined according to the International Classification of Diseases-Tenth Revision codes: ischemic stroke, I63–I639.

The severity of ischemic stroke was assessed by the National Institutes of Health Stroke Scale (NIHSS) [12], which was primarily used to assess the degree of neurological deficit on the day of admission and 2 weeks after onset. NIHSS score < 8 indicates mild stroke (NIHSS score < 4), 8–15 indicates moderate to severe stroke, and more than 15 indicates severe stroke [7]. NIHSS score ≥ 8 is selected as the cutting point.

### 2.6. Statistical Analysis

In the original articles included in our study, there are two types of outcomes that may be related to the purpose of our research; both reported the association between bilirubin and the severity of ischemic stroke. There were two data types for bilirubin as a dependent variable reflecting the relationship with ischemic stroke. Some studies considered bilirubin as a quantitative indicator and reported the mean difference between the two groups (the ischemic stroke severe group vs. the respective control group). Additionally, more studies considered the bilirubin as qualitative indicators that have reported the relative risks (odds ratio (OR), relative risk (RR), and hazard ratio (HR)) with different distributions of bilirubin levels, such as above vs. below the threshold and per tertiles, quartiles, or quintiles in the categorical trait, or the relative risks by per 1-unit increment in the continuous bilirubin traits, which was usually adjusted for other factors. To harmonize the data and facilitate interpretation, we transformed the effect estimates into distributions for the top vs. bottom tertiles and then pooled them [13,14].

We used the Chi-Square-Based-Q-test to assess the heterogeneity among the individual studies. We quantified the heterogeneity with I^2^; if the I^2^ is greater than 50% or if Q statistic had *p* < 0.05 were considered statistically significant. However, even when the heterogeneity was not significant, the results from random effects models (rather than fixed effects models) were reported because of clinical and methodologic heterogeneity, such as differences in study design, the baseline characteristics of patients, and adjustment for confounding variables. We are planning to explore possible causes of substantial heterogeneity with a subgroup analysis (an assessment of heterogeneity). Where there are sufficient data available, we will perform subgroup analyses for the following: the design, the number of samples, and the year of publication. The robustness of the study results will be assessed by a sensitivity analysis, the included studies will be removed one at a time, and the meta-analysis will be performed again to explore the impact of a single study on the aggregate results. The possibility of publication bias was evaluated by the Begg and Egger test, and once bias was apparent, the trim and fill method was applied to make adjustments. The adequacy of the sample size was tested by trial sequential analysis (TSA).

A potentially non-linear dose–response relationship between the bilirubin levels and ischemic stroke severity was modeled using the restricted cubic splines function with four sections on fixed percentiles (5, 35, 65, and 95) of exposure distribution, and a *p*-value for nonlinearity was calculated by testing against the null hypothesis that the coefficient of the second, third spline equaled to 0. If the *p* value > 0.05, the linear model was used for fitting; otherwise, the nonlinear model was used. If the study reported exposure category by a range, the midpoint was calculated by averaging the lower and upper bound; if the lowest category was open-ended, the midpoint was set at half of the upper boundary; when the upper boundary for the highest category was not provided, the midpoint was set at 1.5 times the lower boundary [15]. The Stata software package (version 12.0; Stata company, College Station, TX, USA) was used in our study.

## 3. Results

### 3.1. Search Results

We searched 169 articles from 4 databases. Before screening the literature, 3 duplicates were excluded. Based on title and abstract analysis, 156 studies were excluded (101 articles that did not meet the inclusion criteria after reading the abstracts, 13 animal related studies, and 42 non-original studies); the literature was screened according to the inclusion and exclusion criteria after reading the full text, and we included 7 studies related to the severity of ischemic stroke. A PRISMA flow chart of literature retrieval and selection is shown in Figure 1.

### 3.2. Risk of Bias and Characteristics of Included Studies

The quality scores of the seven included studies based on the NOS assessment tool are shown in Appendix A.

Table 1 showed the relevant characteristics of the included study participants, which included 3 cohort studies [7,8,16], 3 case-control studies [17,18,19], and 1 cross-sectional study [20]. Table 1 also shows the basic characteristics of the study subjects, including the number of subjects in each study (ranging from 73 to 2361), the year of publication (2008–2020), the male proportion (50.90% to 66.31%), age (15–92 years), and information on the underlying diseases of the patients, while other characteristics of included studies are shown in Appendix A.

### 3.3. Correlation between Serum Bilirubin Level and Ischemic Stroke Severity

The research of the association between bilirubin and the severity of ischemic stroke comprised a total of seven studies [7,8,16,17,18,19,20].

We obtained four groups of data from three studies [7,16,20] on the total bilirubin and ischemic stroke severity (NHISS score ≥ 8 points), and the results showed a positive correlation between total bilirubin and stroke severity (OR: 1.14, 95%CI: 1.02–1.26, I^2^ = 68.9%) (shown in Figure 2). Subgroup analysis showed a decreased level of heterogeneity after being grouped according to the number of samples, etc. (Table 2). Additionally in sensitivity analyses, when we omitted each study at a time, the pooled ORs of ischemic stroke severity and total bilirubin levels ranged from 1.12 (95%CI: 1.08–1.15) to 1.16 (95%CI: 1.10–1.23) (Appendix A). In addition, publication bias was assessed by the Begg and Eggger tests, with *p* > 0.05 indicating no significant publication bias (Begg test, *p* = 0.308; Egger test, *p* = 0.626) in the included articles for the correlation between stroke severity and total bilirubin level. The results of subgroup analysis, sensitivity analysis, and publication bias all indicated that our meta-analysis results were stable and credible.

We also pooled the five groups of data of direct bilirubin from four studies [7,8,16,20], and the results showed that direct bilirubin was also positively correlated with the severity of ischemic stroke (OR: 1.53, 95%CI: 1.12–1.94, and I^2^ = 85.1%) (shown in Figure 3). The complete results of subgroup analysis and sensitivity analysis were shown in Table 2 and Appendix A. A publication bias analysis also showed no publication bias (Begg test, *p* = 0.218; Egger test, *p* = 0.211). The reason for the lack of data was that we were unable to pool the relative risks on indirect bilirubin with ischemic stroke severity. 

For direct bilirubin, indirect bilirubin, and total bilirubin, the weighted mean standard deviation (WMD) was not statistically significant in the two groups of severe ischemic stroke (Table 3). 

Due to the lack of data, we could only explore the dose–response relationship between direct bilirubin level and ischemic stroke severity. The results showed that there was a linear relationship between them (*p* _for non-linearity_ = 0.55). Each 1 μmol/L increase in direct bilirubin level may be associated with a 1% increase in moderate or severe ischemic stroke (NHISS score ≥ 8 points) (OR: 1.01, 95%CI: 1.00, 1.01) (shown in Figure 4).

We also compared the first day and 14 days after admission of the differences in the levels of bilirubin (total, direct, and indirect bilirubin); the results showed no significant difference in bilirubin levels in the moderate and severe stroke group (NHISS ≥ 8) at day 1 and day 14 (See Appendix A for the specific results).

### 3.4. Trial Sequential Analysis (TSA)

Trial sequential analysis was performed to verify the adequacy of sample size for the secondary study included. We set 5% of type I error and 80% of power, and we took the sample size as the required information size (RIS). We included seven studies on bilirubin and ischemic stroke severity, of which three studies [17,18,19] met the parameter requirements for TSA. The results showed that the z-curve (blue curve) did not reach RIS (1458), but the z-curve crossed the TSA boundary (the red curve shown in Figure 5) after the inclusion of the second article and the traditional boundary (the red horizontal line shown in Figure 5) after the inclusion of the first article. The TSA results showed that the sample size was sufficient in our study, no matter the relationship between bilirubin and ischemic stroke severity or risk.

## 4. Discussion

Ischemic stroke is defined as a decrease in blood flow to the brain tissue preventing the adequate delivery of oxygen, glucose, and other nutrients, and inflammatory responses caused by reactive oxygen species (ROS), chemokines, and cytokines lead to brain damage [21]. ROS are considered to play a role in the coordinated mechanism of cell signaling and can stimulate many signal transduction pathways that are important in maintaining cellular homeostasis in neurons. Excessive ROS generation can induce the functional and structural damage of neuronal cells through the whole course of AIS, especially the early phase, which may play an important role in the pathophysiology of brain insult [22]. Under normal physiological states, ROS coordination and regulation are controlled by cellular endogenous antioxidants. When this regulation cannot be balanced, oxidative stress ensues [23]. Regarding the oxidative stress biomarkers such as malondialdehyde (MDA), whose concentration was correlated with NIHSS [24], more severe strokes were associated with higher MDA concentration. Different types of circulating bilirubin are effective antioxidants that can react with a variety of reactive oxygen species and have the ability to remove ROS. Several studies have suggested that bilirubin acts as a physiologic antioxidant, with its synthesis being induced in response to oxidative stress. For example, many reports have shown significant increases in serum bilirubin when using halogenated hydrocarbons as oxidative stress inducers [25]. Previous studies have also shown that bilirubin can be associated with the severity of hemorrhagic stroke and brain trauma as an indicator of oxidative stress [26]. The serum bilirubin level in the acute phase of ischemic stroke has not reached a consistent conclusion. Some studies have shown elevated bilirubin levels during the acute phase, suggesting a stress response. We hypothesized that bilirubin may function as an antioxidant and as an oxidative stress biomarker associated with stroke severity. The ultimate goal is to know whether bilirubin can reflect the severity of AIS.

Pineda et al.’s [8] study found a highly significant relationship between ischemic stroke severity and DBIL levels but did not find a significant relationship with TBIL levels. Our results showed that total and direct bilirubin levels were positively correlated with the severity of ischemic stroke. Xu [20] and Luo Yun [7] et al.‘s results showed that higher serum bilirubin was associated with higher NIHSS scores, which was consistent with the conclusions of our study. At the same time, Li [16] et al.’s studies confirmed that triglyceride had a correlation with stroke severity in multivariable logistic regression but had no association in multivariable logistic regression after adjusting DBIL or TBIL, which confirmed that the role triglyceride played in stroke severity was partly mediated by bilirubin. All of the above results support the hypothesis that DBIL and TBIL may be important endogenous antioxidants induced by oxidative stress in ischemic stroke. Like other oxidative stress biomarkers, they reflect the severity of stroke and can be used as an auxiliary indicator to judge the severity of the disease. We also compared changes in bilirubin (total, direct, and indirect bilirubin) levels with the development of acute ischemic stroke from day 1 to day 14. The results at day 14 after hospitalization showed that bilirubin levels tended to decline with the progression of the disease but were not significant, possibly because too few studies were included. However, the decreasing trend suggests that the levels of bilirubin may increase in the acute phase, which may indicate that the increase in bilirubin is caused by the stress response. At the same time, studies have proved that the body can induce the massive synthesis of bilirubin under the stimulation of tissue hypoxia, free radicals, and pro-inflammatory cytokines, and the reduction of antioxidants (such as glutathione) in the body can also promote the increase of bilirubin’s synthesis [27]. In addition, the stress hormone adrenocorticotropic hormone can also increase bilirubin synthesis [28]. We also found a linear correlation between direct bilirubin levels and the severity of ischemic stroke. Studies of various diseases, including liver disease, suggest that DBIL levels may have more prognostic value than TBIL levels, which may also be the reason for the linear relationship between DBIL levels and ischemic stroke severity. 

Our studies have some advantages. First, we analyzed the effects of different types of bilirubin on the severity of ischemic stroke. Second, we explored a dose–response relationship in which bilirubin is linear with the severity of ischemic stroke. Thirdly, besides the sensitivity analysis, the publication bias and TSA showed that the results are stable and the sample size of this study is sufficient, which further enhances the reliability of our conclusion. This study also has several limitations. First, the included studies mainly focused on total and direct bilirubin; there were few studies on indirect bilirubin. Second, our meta-analysis results have some heterogeneity; however, our subgroup analysis, sensitivity analysis, publication bias, and TSA results showed that our conclusions are reliable and stable. Third, the inadequate ORs/RRs under different bilirubin dose points (especially the high bilirubin level) were extracted for our dose–response meta-analysis, so the fitted curve in our study may not be full and accurate enough; a large amount of data is still needed for sufficient curve fitting in the future.

In our study, serum bilirubin levels may reflect the intensity of antioxidant stress in patients with ischemic stroke and can be used as a biomarker to predict stroke severity, which may help determine the state of the disease and also guide the use of drugs. In addition, we need more data from large-sample prospective studies with different types of bilirubin, especially indirect bilirubin.

## 5. Conclusions

In short, the relationship between serum bilirubin and stroke severity meta-analysis suggests that there is a positive correlation between them. Each 1 μmol/L increase in direct bilirubin level may be associated with a 1% increase in moderate or severe ischemic stroke.

## Figures and Tables

**Figure 1 jcm-11-03262-f001:**
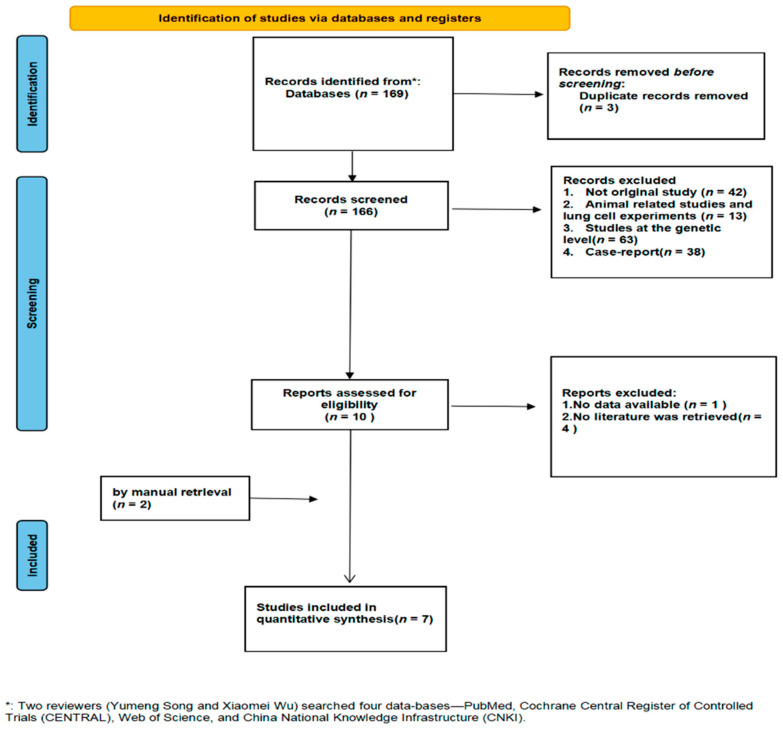
A PRISMA flow chart of literature retrieval and selection.

**Figure 2 jcm-11-03262-f002:**
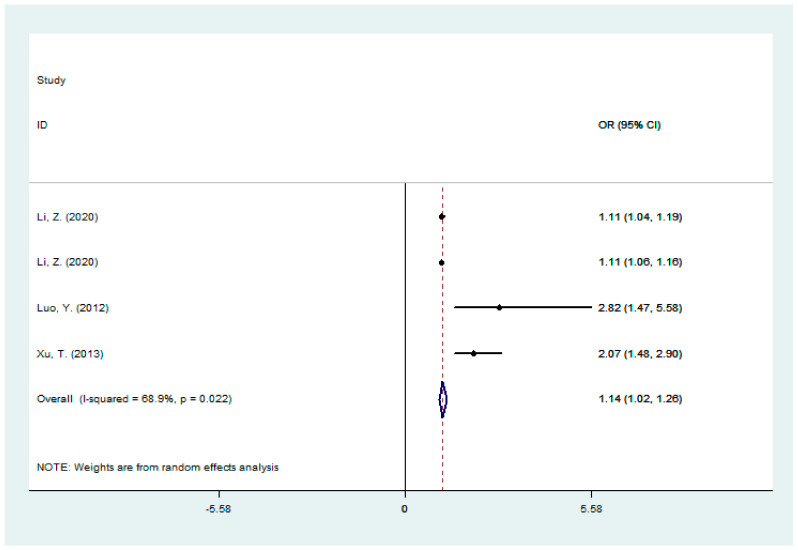
A forest plot of the association between total bilirubin and stroke severity [7,16,20].

**Figure 3 jcm-11-03262-f003:**
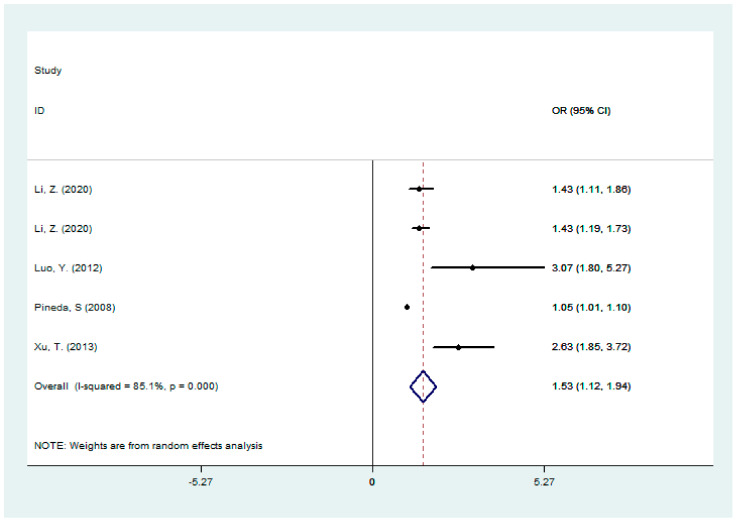
A forest plot of the association between direct bilirubin and stroke severity [7,8,16,20].

**Figure 4 jcm-11-03262-f004:**
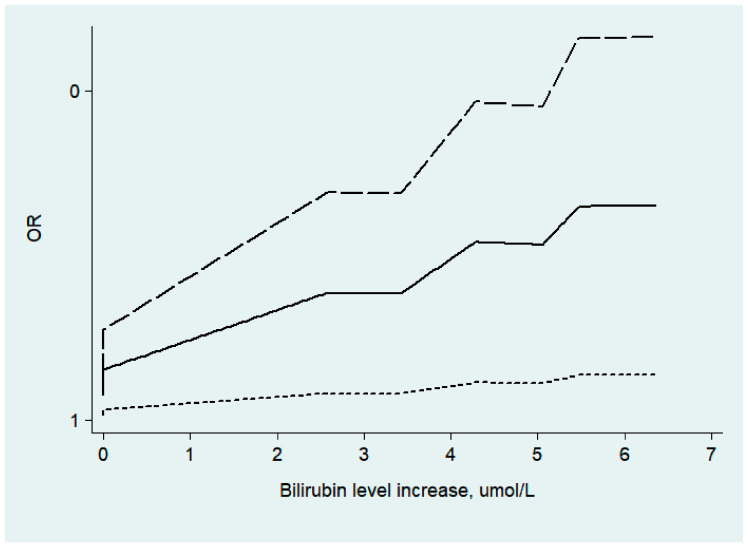
The dose–response relationship between direct bilirubin concentration and ischemic stroke severity.

**Figure 5 jcm-11-03262-f005:**
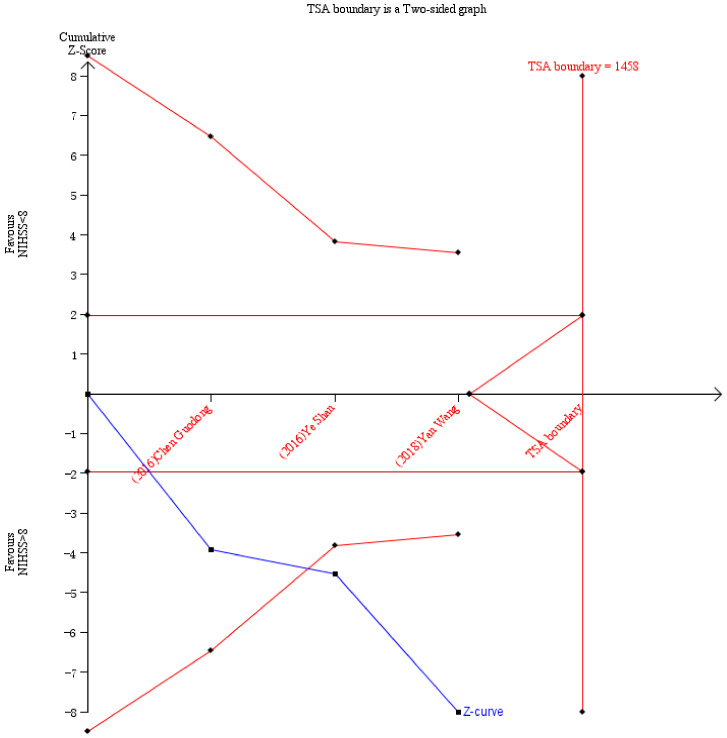
The trial sequential analysis of bilirubin and stroke severity.

**Table 1 jcm-11-03262-t001:** The basic characteristics of the subjects in the included studies.

Study ID	Study Design	Outcome	Age (Average or Range)	Gender (Male%)	Dyslipidemia (%)	Smoker (%)	Alcohol (%)	Hyper-Tension (%)	Diabetes (%)	Atrial Fibrillation (%)	Triglyceride (mmol/L)	Total Cholesterol (mmol/L)	High-Density Lipoprotein Cholesterol (mmol/L)	Low-Density Lipoprotein Cholesterol (mmol/L)
Pineda, S [8]	Cohort	Ischemic stroke	67.5	53.4	N/A	12.7	N/A	67	24.3	19.7	N/A	N/A	N/A	N/A
Luo, Y. [7]	Cohort	Ischemic stroke	15–92	63.4	N/A	N/A	N/A	63.4	N/A	14	1.96 ± 0.46	4.82 ± 0.04	1.14 ± 0.02	2.58 ± 0.03
Xu, T. [20]	Cross-sectional	Ischemic stroke	63.86	63.4	38.1	26.4	N/A	61.4	13.4	3.1	1.54 ± 1.11	5.02 ± 1.18	1.26 ± 0.35	2.99 ± 0.88
Ye Shan [19]	Case-control	Ischemic stroke	55.5	66.31	35.5	25.5	11	69.7	27.9	11.4	1.49 ± 0.81	4.49 ± 1.06	1.04 ± 0.25	3.19 ± 0.94
Chen Guodong [18]	Case-control	Ischemic stroke	62	50.9	N/A	N/A	N/A	60.2	20	7.5	1.37 ± 0.45	5.64 ± 2.21	1.35 ± 1.26	2.55 ± 0.79
Yan Wang [17]	Case-control	large-artery atherosclerotic stroke	58	56.1	11.33	38.4	24.7	87.7	39.7	N/A	N/A	N/A	N/A	N/A
Li, Z [16]	Cohort	Ischemic stroke	66.7	63.1	N/A	N/A	N/A	71.1	33.1	N/A	N/A	N/A	N/A	N/A

**Table 2 jcm-11-03262-t002:** A subgroup analysis of bilirubin and ischemic stroke severity.

	No. of Study Data	OR (95%CI)	I^2^ (%)	*p* for Heterogeneity
**Total bilirubin**				
**Design**				
Cross-sectional	1	2.07 (1.36–2.78)	0	<0.001
Cohort	3	1.11 (1.05–1.17)	24.8	0.265
**Number of samples**				
<1000	3	1.11 (1.05–1.17)	24.8	0.265
≥1000	1	2.07 (1.36–2.78)	0	<0.001
**Year of publication**				
before 2015	2	1.11 (1.07–1.15)	0	1.000
After 2016	2	2.15 (1.48–2.82)	0	0.499
**Direct bilirubin**				
**Design**				
Cross-sectional	1	2.63 (1.70–3.57)	0	<0.001
Cohort	4	1.35 (0.99–1.70)	81.5	0.001
**Number of samples**				
<700	3	1.77 (0.23–3.31)	90.9	0.001
≥700	2	1.49 (1.15–1.83)	40.8	0.186
**Year of publication**				
before 2015	2	2.10 (0.70–3.49)	0	87.6
After 2016	3	1.43 (1.21–1.65)	0	1.000

**Table 3 jcm-11-03262-t003:** The weighted mean difference (WMD) with 95% confidence interval (CI) for the circulating levels of bilirubin between the different groups.

Study ID	Bilirubin (µmol/L), NO.Event	Bilirubin (µmol/L), NO.Non-Event	WMD 95%CI
Total bilirubin			
Chen Guodong	20.85 ± 6.82, 46	16.82 ± 6.21, 62	4.03 (1.53–6.03)
Yan Wang	36.1 ± 2.8, 41	12.2 ± 3.5, 32	23.90 (22.42–25.38)
Ye Shan	16.861 ± 7.689, 80	14.426 ± 6.019, 210	2.44 (0.56–4.31)
Xu, T.	17.97 ± 9.559, 347	14.385 ± 6.926, 2014	3.58 (2.53–4.64)
**Overall**			**8.50 (−2.32–19.32)**
Direct bilirubin			
Chen Guodong	3.84 ± 1.52, 46	3.23 ± 1.25, 62	0.61 (0.12–1.10
Yan Wang	12.8 ± 3.5, 41	3.5 ± 1.2, 32	9.30 (8.115–10.45)
**Overall**			**4.94 (−3.58–13.45)**
Indirect bilirubin			
Chen Guodong	16.28 ± 4.52, 46	14.41 ± 5.22, 62	1.87 (0.03–3.71)
Yan Wang	23.3 ± 5.2, 41	8.7 ± 3.1, 32	14.60 (12.68–16.52)
**Overall**			**8.23 (−4.21–20.71)**

## Data Availability

Not applicable.

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
