# Peer review of "Is Bilirubin Associated with the Severity of Ischemic Stroke? A Dose Response Meta-Analysis"

_jcm, 2022, doi:10.3390/jcm11123262_

Round 1

Reviewer 1 Report

Summary
The authors studied the relationship between bilirubin levels and stroke severity in patients with ischemic stroke. The authors observed that high bilirubin levels are associated with stroke severity.

There are some comments to improve the manuscript as below.

1. Introduction: Please describe the knowledge gap regarding the purpose of this review in detail.

2. Search strategy, page 2: I think the full names of all databases should be included in this section. “Cochrane” and “CNKI” can be revised. In addition, any language or study type restrictions should be described in this section.

3. Study selection, page 2: The eligible study types (e.g., randomized controlled trials [parallel, crossover, cluster], or non-randomized studies) should be described in this section.

4. Outcome definition, page 2: The authors described: “Ischemic stroke was defined according to the International Classification of Diseases Tenth Revision codes: ischemic stroke, I63–I639; hemorrhagic stroke, I60–I629; and all stroke types, I60-I699.” I was a little bit confused. Based on the description, ischemic/hemorrhagic/all stroke were defined as ischemic stroke. Is it right?

5. Statistical analysis, page 3: Please describe the subgroup and sensitivity analyses in detail.

6. Discussion: Please provide the significance and implication of this review in a separate paragraph.

Reviewer 2 Report

This is a good meta-analysis with very standard method and protocol, as well as appropriate structure and scientific writing style. However, there are a few points to be considered. Some of them have herein mentioned:

1- To the best of my knowledge as a neurologist, the "mild or minor stroke" is considered when the NIHSS is less or equal to 4 (and exceptionally, some authors and clinicians have considered 6 as the cut-off point). I do not know the reference for including mild stroke in less than 8 points in NIHSS.

2-Further and more importantly, though, is the fact that it could yield better results if the authors seperated the "moderate" stroke form the "severe" stroke in their clasification and their endpoint due to an array of different pathophysiological and secondary damage of these two categories that mandates their comparison in individual groups.

3- There is not mentioned any data about differences or similarities among the groups of the study regarding other potential confounders such as the lipid profile, diabetes, etc. and any significant statistical difference among them and the method of mitigating such variability or potential error.

4- The article has been written in a fine style and scientific language; yet, it requires minor grammatical or spell checking, for instance, the redundant "c" in "sequenctial" in 123th line.

5- The authors could diminish the similar statements and sentences in their article in comparison to the literatre by paraphrasing or re-writing more sentences or paragraphs.

6- The other shortcoming of this study could be the third "limitation" that the authors have honestly attested in the end part of the discussion.

Overall, I agree that this article is a valuable meta-analysis in the field of "stroke".

Round 2

Reviewer 1 Report

There were no further comments. Thank you.